# Newborn Incubators Do Not Protect from High Noise Levels in the Neonatal Intensive Care Unit and Are Relevant Noise Sources by Themselves

**DOI:** 10.3390/children8080704

**Published:** 2021-08-16

**Authors:** Tanja Restin, Mikael Gaspar, Dirk Bassler, Vartan Kurtcuoglu, Felix Scholkmann, Friederike Barbara Haslbeck

**Affiliations:** 1Department of Neonatology, Newborn Research Zurich, University Hospital Zurich, 8091 Zurich, Switzerland; Dirk.Bassler@usz.ch (D.B.); Felix.Scholkmann@usz.ch (F.S.); friederike.haslbeck@usz.ch (F.B.H.); 2Institute of Physiology, University of Zurich, 8057 Zurich, Switzerland; Mikael.Gaspar@gmail.com (M.G.); Vartan.Kurtcuoglu@uzh.ch (V.K.)

**Keywords:** newborn incubators, neonatal intensive care unit, noise, sound

## Abstract

Background: While meaningful sound exposure has been shown to be important for newborn development, an excess of noise can delay the proper development of the auditory cortex. Aim: The aim of this study was to assess the acoustic environment of a preterm baby in an incubator on a newborn intensive care unit (NICU). Methods: An empty but running incubator (Giraffe Omnibed, GE Healthcare) was used to evaluate the incubator frequency response with 60 measurements. In addition, a full day and night period outside and inside the incubator at the NICU of the University Hospital Zurich was acoustically analyzed. Results: The fan construction inside the incubator generates noise in the frequency range of 1.3–1.5 kHz with a weighted sound pressure level (SPL) of 40.5 dB(A). The construction of the incubator narrows the transmitted frequency spectrum of sound entering the incubator to lower frequencies, but it does not attenuate transient noises such as alarms or opening and closing of cabinet doors substantially. Alarms, as generated by the monitors, the incubator, and additional devices, still pass to the newborn. Conclusions: The incubator does protect only insufficiently from noise coming from the NICUThe transmitted frequency spectrum is changed, limiting the impact of NICU noise on the neonate, but also limiting the neonate’s perception of voices. The incubator, in particular its fan, as well as alarms from patient monitors are major sources of noise. Further optimizations with regard to the sound exposure in the NICU, as well as studies on the role of the incubator as a source and modulator, are needed to meet the preterm infants’ multi-sensory needs.

## 1. Introduction

Newborns born preterm require special care and spend days and sometimes even months in neonatal units equipped with incubators and monitoring tools supporting as well as controlling the infants’ cardiorespiratory function and temperature regulation. These devices contain noise sources, including alarms, switches and fans. Such environmental noise is known to disturb recreation and sleep [1], may lead to long-term hearing loss [2], and has been identified as a potential risk factor for worse neurological development [3]. That environmental noise retards auditory cortical development has been shown in animal experiments [4]. On the other hand, auditory deprivation appears to be a risk factor for unfavorable neurological outcomes [5], while meaningful sound exposure such as music or language may improve the neuronal connectivity [6] and language skills of premature babies [7].

The fetal response to auditory stimuli starts at 19 weeks of pregnancy, enabling the newborn to hear sounds with a frequency up to 500 Hz (for comparison: the most common current concert pitch is 440 Hz). The transmission of frequencies above 500 Hz are limited by the amniotic fluid and the mother’s uterus and surrounding tissue [8]. In parallel with the further development of the neonate, the auditory frequency range increases up to 20 kHz when maturing to term [9]. This maturation process has to take place within the extrauterine environment in the case of premature delivery. New synaptic connectivity has been recently shown to be promoted by acoustic stimuli [6].

Studies demonstrate that preterm newborns may awake in reaction to sound pressure peaks equivalent to 5–10 dB(A) above background noise [1]. Despite this knowledge, unfortunately, little attention has been paid to the auditory surrounding of premature babies. Extrapolation of adult data led the American Association of Pediatrics to recommend to maintain weighted sound pressure levels (SPL) below 45 dB(A) in neonatal intensive care units (NICU) [10]. However, evaluating the noise data in different units demonstrated SPLs that often surpass this threshold [11]. Nevertheless, sound pressure levels alone do not reflect whether a sound is perceived as comforting or disturbing—it might even be difficult to sleep in the presence of a buzzing mosquito, which usually does not elicit sound pressure levels higher than 40 dB(A) [12]. From a structured literature research using the databases Pubmed, EMBASE, and Web of Science in July 2021, 500 reports on “noise and NICU” could be retrieved (Appendix A). These publications demonstrate that noise impacts on newborns, parents and the staff [13,14,15,16,17]. They often either focus on the reduction of noise [13,18] or at the enrichment of the acoustic environment [19,20]. Many studies rely on spot measurements of sound pressure levels and do not consider any frequencies. The incubator itself muffles environmental noise [21] while at the same time exposing the newborn to artificial sounds (e.g., fan noise, door opening and closing sounds). The exact frequency response of an incubator has not been published so far. Here, we therefore aimed to characterize the typical acoustic environment of a preterm baby in an NICU. Our results may be of value for optimizing NICU acoustics to enable optimal development of the neonate.

## 2. Methods

### 2.1. Measurement Setup and Equipment

This study assessed the acoustics inside and outside a running empty incubator (Giraffe Omnibed, GE Healthcare, Ohmeda Medical, Laurel, MD, USA). For all measurements, a calibrated measurement microphone (XREF2, Sonarworks, Riga, Latvia) with a usable bandwidth ranging from 20 to 20 kHz was positioned at the level of the infant’s head inside the incubator 10–15 cm above the mattress at an angle of 30–45° and in the longitudinal axis as illustrated in Figure 1b. To record sound outside the incubator, a second microphone (BCM104, Neumann, Berlin, Germany) with a usable free field bandwidth from 80 Hz to 1.5 kHz was placed at a distance of 20–30 cm from the incubator and 10–15 cm above the mattress. We define the usable bandwidth as the range in which frequency-dependent gain variations are below 2 dB. 

The measurements were performed in three steps: First, the frequency response of the incubator was characterized. Second, different noise sources that commonly occur in an NICU were assessed. Third, the acoustics at our open bay NICU at the University hospital Zurich were directly measured inside and outside the incubator during clinical routine.

### 2.2. Acoustic Properties of the Incubator

In order to characterize the incubator’s frequency response, we performed 60 measurements ranging from 30 s to 2 min in a semi-anechoic room, which was compared to the same measurements in 2 office rooms and inside the hospital. As blankets on top of the incubator are commonly used (Figure 1a), the effect of this measure was evaluated as well.

### 2.3. Transfer of Noise into the Incubator

Using the same equipment described above, we measured the noise generated by regular activities such as opening and closing the access doors of the incubator, opening the lid or refilling the water tank. Additionally, the transfer of monitor alarms as generated by the monitoring system (IntelliVue MX550, Philips, Amsterdam, The Netherlands) were measured.

### 2.4. Characterization of the Environmental Noise at the NICU 

To assess for the acoustic environment at our NICU during clinical routine procedures, we evaluated intermittently 35 h and 14 min comprising five random time points, finally covering a full day and night period outside and inside the incubator positioned at the NICU at the Department of Neonatology at the University Hospital in Zurich, Switzerland. This NICU has an open bay design (Figure 1a).

### 2.5. Data Recording and Processing 

Data were recorded uncompressed in Waveform Audio File format at a bit depth of 16 bit and with a sampling rate of at least 48 kHz. Data processing was performed in Matlab R2019a (Mathworks, Natick, MA, USA).

## 3. Results

### 3.1. Acoustic Properties of the Incubator

After the start-up procedure, the incubator produces noise in specific frequency bands, i.e., a very low-frequency noise in the range of 20–30 Hz, a 100 Hz hum (most likely attributable to the transformer) and two constant high-frequency noise components in the range of 1.3–1.7 kHz as well as at 3 kHz. Inside the incubator, the SPL was 34.7 ± 0.5 dB(A) when the incubator was switched off and 40.5 ± 0.5 dB(A) after the device completed the start-up procedure. The characteristic spectral signatures of the incubator are displayed in Figure 2a. The origin of the detected noise within a frequency range of 1.3–1.5 kHz with weighted SPL of 40.5 dB(A) is the incubator’s fan. This ventilation system masks sounds originating outside the incubator with SPL less than 50–55 dB(A). Additionally, the attenuation of sound passing into the incubator is frequency dependent, which is further discussed below. With the doors closed, the incubator’s architecture and material did not induce any dampening of sound or noise at frequencies below 250 Hz. Above 250 Hz, SPL was reduced by 15 dB on average. The incubator properties with respect to sound transmission from the outside to the inside were identical no matter whether the sound transfer was measured in the semi-anechoic room or in the NICU, as displayed in Figure 3a. The use of a blanket on top of the incubator provided only little benefit in reducing sound transmission above 2 kHz. Modification of the amplitude spectrum in the incubator was dependent on the amount of area covered by the blanket.

### 3.2. Transfer of Noise into the Incubator

The noise inside the incubator has a strong low-frequency component (at about 20–250 Hz) with a larger amplitude than the noise in the NICU. Sound at frequencies above approximately 300 Hz is attenuated markedly as it passes into the incubator. The temporal characteristics of noise inside and outside the incubator differ (Figure 3b,c), with the inside showing a narrow range of sound pressure levels (SPL: 40.4 ± 2.2 dB(A)) than the outside (SPL: 51.2 ± 5.6 dB(A)). Opening and closing of one door added 30 dB(A), while the closing of one door while another was open led to a 15 dB(A) increase in SPL. Opening the incubator on one side, as done for X-ray evaluation, causes an SPL increase of 15–20 dB(A) and closing it of 38–42 dB(A), both during less than 0.4 s. We further found that the incubator water tank is an additional unexpected source of the noise. Its closure causes a transient sound of approximately 0.3 s duration with a sound pressure level exceeding 70 dB(A). Opening and closing the top (as done for catheter insertion or complex intubation procedures) causes sound peaks beyond 70 dB for approximately 0.5 s and 0.3 s, respectively.

Placing a plastic box on top of the incubator, a procedure not recommended but commonly discussed in the literature or observed (positioning of disinfectant or injection cannulas) yielded events above 80 dB(A), lasting for periods less than 0.3 s. The Philips IntelliVue MX550 monitor positioned next to each incubator generates a warning and a danger sound depending on the circulatory monitoring of the patient. Warning sounds were measured at 480 Hz (blue and orange alarm) and danger sounds were measured at 960 Hz (for the red alarm). Modes above harmonics are visible in the spectrogram up to 11 kHz (Figure 2b,c).

### 3.3. Characterization of the Environmental Noise at the NICU

The average weighted SPL during acoustic evaluation at the NICU measured next to the incubator was 53 dB(A). During more than half (54%) of the whole measurement time at the NICU it was above 45 dB(A). During the recordings at the NICU outside the incubator, we detected 194 occurrences of weighted SPL exceeding 65 dB(A), most of which were of short duration (<0.4 s) and attributed to the opening and closing of a cabinet at the entrance of the NICU. 

## 4. Discussion, Conclusions, and Outlook

### 4.1. Acoustic Properties of the Incubator

Our data demonstrate that despite the progress made in incubator technologies reducing the noise SPL from 70–80 dB(A) to 44 dB(A) within the last 40 years, newborns within an incubator in the NICU are still exposed to high noise levels [22,23,24]. Even in the absence of additional noise from sources other than the incubator fan, the preterm environment would not meet the recommendations of the World Health Organization for community noise, which state that continuous background noise levels should not exceed 35 dB(A) during sleep and individual noise events should not exceed 45 dB(A) [25]. Since then, these standards have been repetitively integrated in the current recommendations for environmental standards in NICUs [26,27,28,29]. The frequency range of the measured constant fan noise, i.e., 1.3–1.5 kHz, partially overlaps with the frequencies covering the range of human speech [30]. Although the audible range lies between 20 Hz and 20 kHz in humans, speech can be regularly identified within the more narrow band width between 300 Hz and 3.4 kHz, which therefore has been used as the “telephone bandwidth” according to the standards of the International Telecommunication Union [31,32]

Additionally, frequency-dependent attenuation changes the amplitude spectrum of speech as the corresponding sounds enter the incubator. According to Fletcher et al. (1923), the intelligibility of speech depends on the sum of different frequency bands, speech intensity, and temporal properties of the stimulus [33,34]. Consequently, the reduction of speech loudness by 15 dB(A) at frequencies above 250 Hz will impair the transfer of speech into the incubator. If a person talks to a newborn inside the incubator, the incubator-related reduction of the spectral bandwidth will interfere with speech intelligibility. As a result, sound transfer will be impaired, which is in line with the findings of French et al. [35]. Therefore, if a caregiver wants to read or sing to the child, the side port of the incubator should be opened. Additionally, the lower frequencies will be better audible than higher, which might explain why in a recent analysis the exposure to mothers’ voices led inside an incubator induced less relaxing physiological reactions in the newborns compared to white noise exposure [36]. 

### 4.2. Transfer of Noise into the Incubator

However, although each incubator does not produce sound exceeding 45 dB(A), it is not taken into account that the aggregation of different machines in the NICU and the effect of further noise sources, such as an incubator port or cabinet door openings, cumulate together and produce a noisy atmosphere, masking meaningful speech and sound directed toward the newborn. In addition, the transfer of respirator tubings and cables via the small open side ports may generate noise, especially if the tubings have an irregular surface as is the case with accordion-tube design. These noise intensities will differ depending on how gently the cables are handled. The fact that opening of the first and closing of the last incubator entry port multiplied the loudness by more than the factor of 10 (35 dB(A)) can be attributed to the lack of a decompression system. Healthy newborns have shown to wake up in response to 3 min of mixed noise exposure between 100 Hz and 7 kHz at sound levels of 70–75 dB [37]. However, it is likely that preterm and sick patients are more vulnerable [38]. 

Our data show that preterm infants are exposed to arguably high and diffuse non-contingent auditory overstimulation. Cotton covers may help against light, but they do not considerably reduced infant noise exposure in contrast to more noise absorbing polyurethane foam panels evaluated by Bellieni et al. [39]. Nonetheless, better insulating panels do not protect from fan noise that originates inside the incubator [39]. There is only one randomized controlled trial to reduce noise so far. However, in this small newborn cohort, silicone ear plugs may have been effective in the reduction of noise exposure [13,40]. However, they will not protect against respirator-associated noise transferred via bone conduction and will not help in distinguishing noise from sound, either. If monitor providers would predominantly use alarms at higher frequencies, the corresponding sounds would be attenuated more strongly as they pass into the incubator. However, when lying in the warmer or the arms of the parents, alarms would still disturb the infants, families, and NICU staff, so that visual or sensory alarms might be an alternative option to reduce noise [41]. In order to ensure persistent noise reduction, repetitive training and increased awareness of the personnel is very important; otherwise, the effect of improvements will get lost [42]. 

### 4.3. Characterization of Environmental Noise at the NICU

Empirically, based on our own observations, the number of noisy technical devices, the patient turnover, and visiting times of relatives and personnel has increased during the last several years. All these procedures may contribute to the increased background noise measured in our NICU. Correspondingly, Busch-Vishniak et al. [43] found “a clear trend for rising hospital noise levels” since 1960 at a rate of 0.38 dB per year for daytime levels and of 0.42 dB per year for nighttime levels. NICU noise levels are still reported to range from 54 to 60 dB(A) [44], reaching peaks of 120 dB and exceeding recommended sound levels more than 70% of the time [45]. Berg et al. [46] underline these observations demonstrating peak noise levels between 82 and 102 dB(A), particularly during visiting hours and medical rounds, and peak noise levels around 75 dB originating in alarms and other technical devices. In their systematic reviews, O’ Callaghan [47] and Veenendaal et al. [48] argue for an “evidence-based design” of NICUs, where single family rooms seem to have advantages concerning the environmental noise control [49,50,51,52,53,54]. One issue that is beyond the scope of our analysis is the noise generation by different systems used for newborn respiratory support. Ventilators for mechanical ventilation and non-invasive respiratory support systems both generate significant sound pressure levels up to 100 dB when measured in the postnasal space [55], depending on the device characteristics [56] and on the flow rate and amplitudes that are used [55]. Unlike external noise, which may be reduced by protective devices, there is nearly no attenuation of sound transfer via bone conductance [57]. Surenthiran also measured the in-the-ear noise intensities at 1 kHz, which showed a mean noise of 55 dB SPL if 5 L flow per minute was used with a device for continuous positive airway pressure [55]. To our knowledge, there is no qualitative measurement of NICU environmental sound that might distinguish disturbing, irregular noise from meaningful sound exposure. Interestingly, the same sound may be interpreted as comforting sound or annoying noise depending on the individual situation [58,59], associated expectations and interpretation [6,60], and the cultural background [61]. Some people have shown a higher noise sensitivity [62], with an estimated hereditability of about 30% [63]. That is why intensive care should become more personalized and humane to generate a positive basic atmosphere [64]. 

Therefore, the assessment of what is comfortable and what is not for an individual patient can only be based on close observation of the preterm. In contrast to adults where the sleeping time concentrates on several hours during the night, newborns have irregular sleeping patterns without an established circadian rhythm [65], which makes the organization of an open ward even more challenging. In order to adapt the acoustical environment to each patient’s needs, we would need a flexible acoustic environment in each patient zone, which could be realized with advanced acoustic curtains. Since family-centered care increases interaction between parents and staff, the communication has to be properly dosed in order to provide rest and recreation for neighboring families. A recent review of intensive care unit built environments addresses this problem [15]. Studies suggest that parental vocal qualities are commonly adapted to their infants’ behavioral state [66], which is why an attenuation of either the infants’ state and/or the parental voice by the noisy environment and/or incubator possibly impairs proper interaction. Current monitor alarms produce sound at or above 480 Hz, which is within the frequency range of newborn babies’ cries [67]. These characteristics are certainly useful to attract the staff’s attention, but they may impair newborn and parental comfort or rest. 

### 4.4. Strengths and Limitations of This Study

This study characterizes the acoustic properties of the Giraffe Omnibed (GE Healthcare, Ohmeda Medical, Laurel, MD, USA) incubator. Since the retrieved data concerning the sound pressure level inside the incubator and the frequency response have been shown to be reproducible in a semi-anechoic room, different offices, and the NICU, the data that are presented seem to be robust and transferable to similar incubator models as well. The bit depth of 16 bit and a sampling rate of at least 48 kHz is far higher than in most studies where there were no continuous measurements. This approach enables us to estimate the acoustic environment of a newborn inside the incubator. However, the microphones that were used for recordings limited the frequency spectra, which we could analyze to a bandwidth between 80 Hz and 1.5 kHz, not taking into account higher or lower frequencies that still might play a role. Additionally, the acoustic properties of the architecture at our NICU have not been analyzed, and the modification of sound generated by the newborn inside the incubator has not been studied. With the measurement of frequencies and sound pressure levels, we cannot distinguish between meaningful sound exposure and disturbing noise, and we did not perform any long-term measurements. This study offers several approaches to reduce noise such as taking into account potential noise sources (i.e., cabinet doors, incubator ports, visiting rounds); however, we have not developed any concrete solution to this problem. The evaluation of long-term effects of acoustic improvements on the baby would be the most interesting data, which is still open for further investigation. 

### 4.5. Conclusions for the Newborn inside the Incubator

We assume that with a baby inside, especially if the baby is sick, repetitive opening and closing of the incubator ports will be the most important noise source. If a baby does not have to be accessed, opening and closing of the cabinet doors or drawers were the most important source of mechanical, loud, unpredictable (>65 dB), and short sudden (<0.4 s) noise. Although there may be interindividual, situational, and cultural differences with respect to the definition of a pleasant or unpleasant acoustic surrounding, there is still a consensus that humans generally prefer clear rather than distorted sounds [68]. Our findings are in line with similar studies that identified a noise generation between 80 and 90 dB(A) due to the incubator opening or closing, which have been reviewed extensively [69,70]. Unfortunately, these findings do not comply with the recommendations to maintain the combination of background and operational sound within an hourly equivalent continuous sound pressure level (SPL, Leq) of 50 dB, referring to a weighted slow response. Moreover, the SPL of 55 dB should not be exceeded more than 10% of the time (l 10). Opening the incubator at one port does not comply with the recommendation that transient sounds shall not exceed 70 dB. Depending on the attention of the listener, the amount of vowels in the input sound, and the frequency spectrum which is used, speech has shown to be intelligible if the sound pressure level is about 9–18 dB [71] higher than the background sound pressure level and can be predicted according to the international standard IEC 60,268 [72]. Structured, repetitive prosodic patterns seem to be most important for language learning [73]. However, the acoustic environment of preterm infants is rather characterized by unstructured, sudden and unpredictable noise sources. Knowing that at 1 kHz and above 40 dB, the perceived loudness doubles if the sound pressure level is increased by 9 dB [33], it visualizes that the loudness perceived by preterm infants increases substantially by minor manipulations such as accessing the baby, by alarms, or cabinet use. This is because perceived loudness roughly doubles with each 10 dB increase in SPL (more accurately, it doubles with each 9 dB increase in SPL at 1 kHz and above 40 dB). Children need a higher signal-to-noise ratio to perceive speech correctly [74], making it likely that precisely articulated meaningful sound exposure such as with “motherese” or “infant-directed speech” [75] is necessary for the neurodevelopment of newborns as well. However, the evaluation of Caskey et al. [7] revealed that language exposure only contributed for up to 5 ± 3% of sound recordings at an NICU, which increased with gestational age. This means that a premature baby in a closed incubator is largely isolated from conversations in the NICU but not from loud noises produced in the unit. In our unit, the parents normally come for 1–3 h a day to care for their child. On average, they might talk or sing to their baby for about 1 h daily. Assuming 12 caring procedures per day in the extremely premature, the staff will probably additionally talk for 5 min during each of these procedures. Altogether, this might lead to a meaningful exposure to sound during 8% of the daytime. The rest of the day, they will experience a mixture of sound and noise with both background noise, technical alarms, and medical discussions. Interestingly, EEG data deriving out of adult patients in the ICU suggest that REM sleep is most severely affected by the ICU surrounding [16]. Since the percentage of REM sleep is significantly higher in newborns compared to adults, this finding of sleep disturbance raises additional concern. When compared to other NICUs, our open bay NICU with an average SPL of 53 dB(A) does not seem to be a loud one; the staff is trained to lower their voices, we offer music therapy, and we encourage parents to communicate with their newborns. Correspondingly, in a recent study in Montreal even after change of the whole NICU architecture, an average sound pressure level of 49 dB(A) (coming from 58 dB(A)) has been reported [76]. Similar ranges have been reported in South India [77]. 

Depending on the child, some nurses observed that during the most common visiting hours such as during the weekends or if several urgency admissions took place, their newborns showed a higher rate of apnea and higher arousal frequencies. 

Considering that newborn exposure to meaningful sounds contributes to their language acquisition and is essential for newborn development [78], our findings raise concerns and demand improvements in incubator technology and acoustic architecture at NICUs. Both human factors (staff and families), direct (incubator) and indirect infant and personnel surroundings (building properties, drawers, wardrobes), as well as technical equipment (ventilators, monitoring) have to be considered. The staff and families need continuous education to reduce their sound pressure levels during interactive speech but to increase the levels and open the incubator ports if the talk is directed to the newborns. While direct speech provides a very broad frequency range between 20 Hz and 20 kHz, microphones and loud speakers display a spectral narrowing (displayed by respective provider). Alarm levels should be reduced, and incubator ports and drawers have to be opened gently. Some background noise will always be audible, but it will be interpreted differently depending on the infant’s health state and genetic background. Consequently, effort should be made both to monitor and to decrease noise sources, for example via portable applications [79,80]. In addition, meaningful acoustic stimulation of preterm babies in an NICU should be increased, providing vibroacoustic enrichment by meaningful auditory stimulation and social contact with infant-directed music and parental empowerment to speak and sing for their infants [6,81]. Offering zones where staff and/or parents can communicate and discuss with each other without disturbing the infants sleep is of great interest. Moreover, the definition of a “good acoustic environment” through only a weighted sound pressure level threshold should be revised, taking into account the quality of the sound. The ultimate goal will be to balance sounds and shape the preterm acoustic environment according to the infants’ multi-sensory and social needs.

## Figures and Tables

**Figure 1 children-08-00704-f001:**
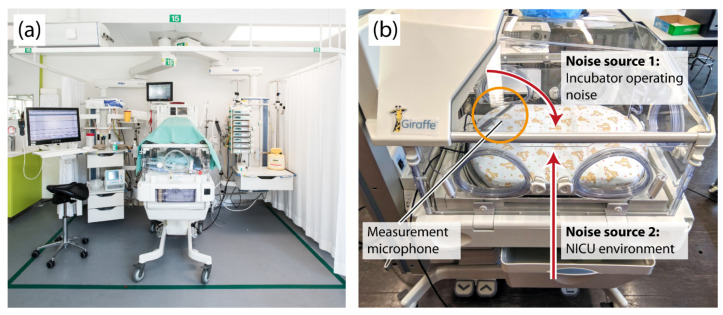
(**a**) Incubator in an NICU. In the picture, half of the top of the incubator is covered by a blanket to decrease the brightness inside the incubator. Image source: University Hospital Zurich; with permission. (**b**) Close-up of the incubator used in the study. The paths of two main sources of noise for the incubator are indicated by red arrows: sound coming from outside and inside. The orange circle indicates the position of the measurement microphone.

**Figure 2 children-08-00704-f002:**
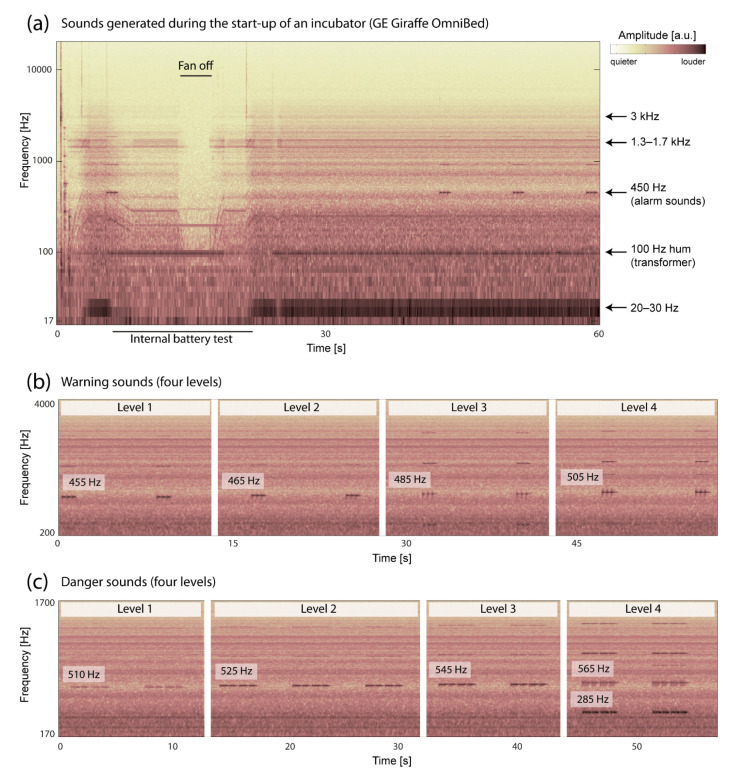
(**a**) Spectrogram of the noise produced by the incubator during the start-up phase. (**b**,**c**) Spectrograms of the warning and danger sounds produced by the IntelliVue MX550, Philips monitor next to the incubator.

**Figure 3 children-08-00704-f003:**
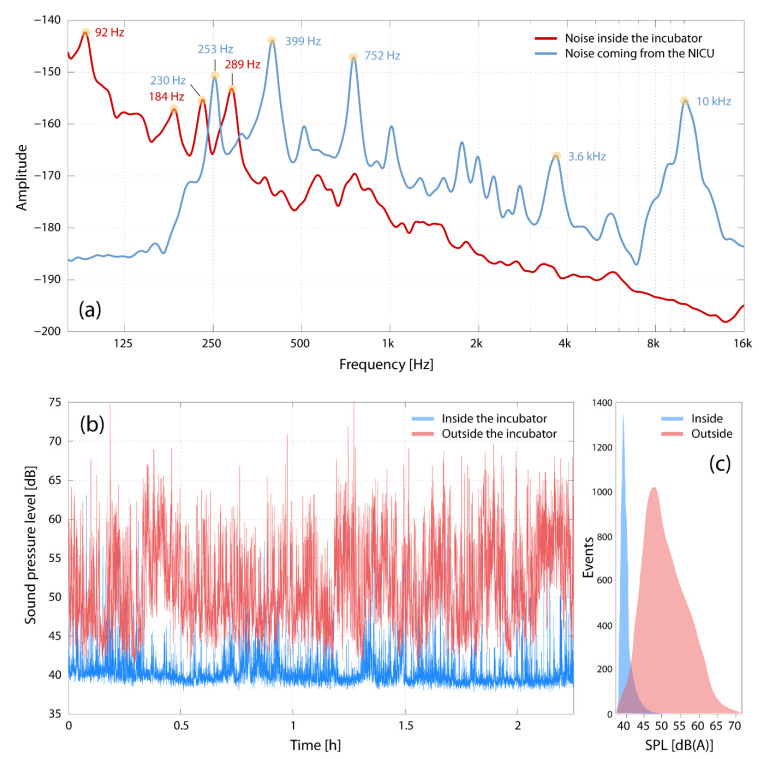
(**a**) Noise spectra recorded inside and outside an incubator located in an NICU. Peak frequencies are indicated at the specific peaks. The recording was done in an NICU with several incubators working and under normal clinical working conditions. (**b**) Time-series of SPL variations over about 2 h, measuring inside and outside an incubator. (**c**) Distribution of SPL values shown in (**b**).

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
