# Peer review of "Newborn Incubators Do Not Protect from High Noise Levels in the Neonatal Intensive Care Unit and Are Relevant Noise Sources by Themselves"

_children, 2021, doi:10.3390/children8080704_

Round 1

Reviewer 1 Report

Brief summary:

This is a nice article on sound levels in a neonatal intensive care unit (NICU) in a single university hospital in Switzerland. The authors describe a typical acoustic environment for a preterm infant in an incubator. It is a very nice article with relatively detailed measurements of various environmental stimuli inside and outside the incubator that are transmitted into the incubator. I do also like the chapter where the authors comment on strengths and weaknesses of their study. A few comments I would like to make and would like the authors to address before recommending this for a publication:

Comments:

  1. Authors measured noise levels in an open bay NICU. I would like authors to comment on any available studies regarding noise levels in an open bay NICU compared to a single-room NICU. This would help the reader extrapolate and understand their data in relation to their NICU.
  2. Regarding “a running empty incubator” authors used to measure sound levels. Were devices such as tubing systems for mechanical ventilator connected to the incubator? Having this tubing go through the holes of the incubator may increase area from which noise comes into an incubator.
  3. Regarding “running an empty incubator” – 2nd question: Did the authors make any measurements with regards to noise levels of mechanical ventilators (conventional vs oscillator). I would like authors to provide this information. A large proportion of premature infants are intubated at some point, some only short period of time, some much longer. It would be important to know/understand the noise levels that a mechanical ventilator (conventional and/or oscillator, or both) cause as this is likely a major source of toxic noise stimulus.

If above comments/suggestions are considered (broad and specific comments), I think this study will add to the field and be welcomed by the readers of Children.

Author Response

Dear reviewer,

thank you for taking your time to read our manuscript and for your valuable suggestions. Please find the responses to your questions attached. Additionally, we now could additionally provide the data on 5 days and 5 nights of qualitative assessment, if you think this would add important information. 

Best regards, 

Tanja

Reviewer 1

  1. Authors measured noise levels in an open bay NICU. I would like authors to comment on any available studies regarding noise levels in an open bay NICU compared to a single-room NICU. This would help the reader extrapolate and understand their data in relation to their NICU.

Thank you for this valuable input. According to your suggestion we added the explicit information of the open bay design in line 109: “This NICU has an open bay design (Figure 1a). “

Moreover, we discuss the potential advantages of this NICU design in line 246ff: “In their systematic reviews O’ Callaghan [1] and Veenendaal et al. [2] argue for an “evidenced-based design” of NICUs, where single family rooms seem to have advantages concerning the environmental noise control [3-8].”  

  1. Regarding “a running empty incubator” authors used to measure sound levels. Were devices such as tubing systems for mechanical ventilator connected to the incubator? Having this tubing go through the holes of the incubator may increase area from which noise comes into an incubator.

This is an interesting question which we considered as well. The different tubing systems vary a lot and most of them will be tightly sealed due to the rubber closure at the edges of the incubator and, therefore, should not modify the internal sound characteristics considerably. However, the transfer of a respiratory tubing, especially those with a rough accordion-tube design through the small open ports generate fairly high levels of noise, but the final intensity will depend on whether it is transferred gently or not. We now mention these considerations in line 210: In addition, the transfer of respirator tubings and cables via the small open side ports may generate noise, especially if the tubings have an irregular surface as is the case with accordion-tube design. These noise intensities will differ depending on how gently the cables are handled.“

  1. Regarding “running an empty incubator” – 2nd question: Did the authors make any measurements with regards to noise levels of mechanical ventilators (conventional vs oscillator). I would like authors to provide this information. A large proportion of premature infants are intubated at some point, some only short period of time, some much longer. It would be important to know/understand the noise levels that a mechanical ventilator (conventional and/or oscillator, or both) cause as this is likely a major source of toxic noise stimulus.

Thank you for this input. We, indeed, first considered to measure with different mechanical ventilators, however, we have different respirators in use and the noise seems to vary upon the ventilatory adjustments with different pressure and flow levels, which are applied. This issue which leads to considerable noise levels via bone conductance has been addressed previously by  Gol et et al. We therefore now add the following section in line 246ff.: “One issue which is beyond the scope of our analysis is the noise generation by different systems  used for newborn respiratory support. Ventilators for mechanical ventilation and non- invasive respiratory support systems both generate significant sound pressure levels up to 100 dB when measured in the postnasal space [9], depending on the device characteristics [10] and on the flow rate and amplitudes which are used [9]. Unlike external noise which may be reduced by protective devices, there is nearly no attenuation of sound transfer via bone conductance [11]. Surenthiran also measured the in-the-ear noise intensities at 1 kHz which showed a mean noise of 55dB SPL if 5l flow per minute was used with a device for continuous positive airway pressure [9]

Reviewer 2 Report

Review Children - 1254575

Dear authors, thank you for submitting this manuscript to Children. The article is well written and appropriately structured. Thus, I believe it may be suitable for publication after major changes. Please consider the suggestions below before re-submitting the manuscript. Congratulations.

  • On p.6. you state: “The frequency range of the measured constant fan noise, i.e. 1.3–1.5 kHz, overlaps with the frequency range of human speech [15].”

Please clarify, since I believe human speech has a much lower frequency range?

  • You acknowledge the limitation in not distinguishing between disturbing sounds (e.g. noise) and meaningful sounds (e.g. speech) in you study. Anyways, it would be enriching to have an estimate of the components of the overall SPL, e.g. noise vs speech, sound levels of staff talking vs. technological sounds. Otherwise, it will be difficult to draw conclusions.
  • Both meaningful and disturbing sounds and noise have universal, cultural and biographical significance. I would suggest you acknowledge that your study took place in a specific cultural region, where sensitivity to and expression of sounds are also shaped by cultural and societal norms. This does not only refer to human speech, but also to what is perceived as ‘loud’ or as ‘noisy’. While 55-60 db(A) might be perceived as noisy in Switzerland, this might be just a common background sound level in other countries and regions of the world. Are there any studies that show differences of babies in terms of sensitivity to sound depending on cultural and geographical regions?
  • Besides that you state that the sound levels in your NICU are above the recommended levels, is this of clinical relevance for your population? For example, did you observe alterations of infants' behavior, physiology, development that result from exceeded sound levels?
  • On page 8 you state: “Consequently, effort should be made to increase meaningful acoustic stimulation of preterm babies in an NICU, e.g., with Creative Music Therapy providing vibroacoustic enrichment by meaningful auditory stimulation and social contact with infant-directed music and parental empowerment to speak with “motherese” hum and sing for their infants [6,36].”

This is the first time you mention Music Therapy in the article, and the reader might not be familiar with it. Please explain briefly. You mention Creative Music Therapy as meaningful stimulation. Why do you specifically address CMT and not MT in general as meaningful stimulation? I would suggest justifying just mentioning CMT or leave it more general, since I believe this article is not specifically about CMT?

  • You state that specific aspects of the incubator (the fan, opening doors) exceed suggested sound levels. However, in the APA guidelines, peak sound levels for short periods of time are considered. You mention such guidelines briefly in the introduction section, but I would suggest coming back to them also in the discussion. In the discussion you mention the WHO guidelines for community noise, which is probably not the most appropriate reference since the NICU is a specialized setting. Besides, the reference is from 1999 and is most likely not up to date. 
  • In the discussion there is a brief section on environmental noise in the NICU, but the references you mention are all from between 10 and 15 years ago. There are many new studies on the sound environment in the NICU, I would suggest updating your information.
  • There is no section in the article that refers to a literature review. Besides that it is known that noise can alter preterm infants sleep, physiology and development, what are the sound levels in other units and countries? Here are just some examples:
    • doi: 0.1016/j.ijporl.2018.02.013
    • doi: 10.1111/apa.13906

I would suggest providing a more thorough literature review at the beginning of the article so the reader gets an idea of what is known already and can appreciate your study within a wider field of knowledge. Also, you should come back in the discussion and compare your results to other studies. What is unique about your study? How does your study help to further advance knowledge in the field? This should be addressed.

  • There are no recommendations on what to do with the information obtained in your study besides referring to structural (NICU) and manufactural (incubator) improvements that should take place. What do your results mean for parents, staff, music therapists? What recommendation can you make? Should music therapists or parents open the doors of the incubators when talking or singing? Should they be closed? Are there any differences for voice vs instruments? How can staff use this information to better protect preterm babies? It is essential to extrapolate your findings to provide concrete information for relevant stakeholders in the NICU.

Author Response

Dear reviewer, 

thank you very much for your valuable suggestions which we tried to answer appropriately. If you think it is useful we can offer to add a qualitative assessment of 5 days and 5 nights, where we tried to differentiate the different sound sources. 

Best regards, 
Tanja 

Reviewer 2:

Dear authors, thank you for submitting this manuscript to Children. The article is well written and appropriately structured. Thus, I believe it may be suitable for publication after major changes. Please consider the suggestions below before re-submitting the manuscript. Congratulations.

  • On p.6. you state: “The frequency range of the measured constant fan noise, i.e. 1.3–1.5 kHz, overlaps with the frequency range of human speech [15].”

Please clarify, since I believe human speech has a much lower frequency range?

This cited statement within the manuscript is scientifically correct. To avoid misunderstandings we adapted it accordingly: 186ff: “The frequency range of the measured constant fan noise, i.e. 1.3–1.5 kHz, partially overlaps with the frequency range of human speech [15]. “

“Although vowels are of lower frequencies, the intelligibility of speech seems only be appropriate if the audibility within a frequency range between 300Hz-3.4kHz is guaranteed, that is why the “telephone bandwidth” has been defined accordingly. In order to clarify the background, we added the following section in line 188ff: “Although the audible range lies between 20Hz to 20kHz in humans, speech can be regularly identified within the more narrow band width between 300Hz–3.4kHz which has therefore been used as the “telephone bandwidth” according to the standards of the International Telecommunication Union [12,13].”

  • You acknowledge the limitation in not distinguishing between disturbing sounds (e.g. noise) and meaningful sounds (e.g. speech) in you study. Anyways, it would be enriching to have an estimate of the components of the overall SPL, e.g. noise vs speech, sound levels of staff talking vs. technological sounds. Otherwise, it will be difficult to draw conclusions.

Thank you for your input. We indeed tried to differentiate the different sounds and noise sources and rate them. However, due to the open bay architecture, there is nearly always a mixture of both technical alarms and artificial noises (ventilators, monitors, opening of drawers) together with speech. In order to give an impression of the situation at our ward, we now added the following lines in 334 ff: In our unit, the parents normally come for 1-3 hours a day to care for their child. On average, they might talk or sing to their baby for about 1 hour daily. Assuming 12 caring procedures per day in the extremely premature, the staff will probably additionally talk 5 minutes during each of these procedures. Altogether, this might lead to a meaningful exposure to sound during 8% of the daytime. The rest of the day, they will experience a  mixture of sound and noise with both background noise, technical alarms and medical discussions. Interestingly, EEG data deriving out of adult patients in the ICU suggest that REM sleep is most severely affected by the ICU surrounding [14]. Because the percentage of REM sleep is significantly higher in newborns compared to adults, this finding of sleep disturbance rises additional concern.

In order to further differentiate between technical noise and talk, Servagathasenay Yohasenan assessed 6 days and 5 nights in our NICU and rated the quality of sound. However, she found out that the impression of how loud a certain noise was depended on the background noise and her own activity and health state. If you think this might be informative for the reader we can provide this qualitative approach. In order to assess the amount of personnel at our NICU we counted the staff entering the NICU. If you consider this information to be valuable, we can add these graphs and would add her as a co-author our seven day deadline was not enough to complete this evaluation.

Both meaningful and disturbing sounds and noise have universal, cultural and biographical significance. I would suggest you acknowledge that your study took place in a specific cultural region, where sensitivity to and expression of sounds are also shaped by cultural and societal norms. This does not only refer to human speech, but also to what is perceived as ‘loud’ or as ‘noisy’. While 55-60 db(A) might be perceived as noisy in Switzerland, this might be just a common background sound level in other countries and regions of the world. Are there any studies that show differences of babies in terms of sensitivity to sound depending on cultural and geographical regions?

We agree that there is a universal, cultural and biographical significance of sound. Consequently, we now added line 259ff.: “Interestingly, the same sound may be interpreted as comfortable sound or annoying noise depending on the individual situation [15,16], the expectancy and interpretation [17,18], and the cultural background [19]. Some people have shown a higher noise sensitivity [20], with an estimated hereditability of about 30% [21]. In addition we write in line 277: Although there may be inter-individual, situational and cultural differences with respect to the definition of a pleasant or unpleasant acoustic surrounding, there is still a consensus that humans seem to prefer rather smooth than rough tones [22].

Besides that you state that the sound levels in your NICU are above the recommended levels, is this of clinical relevance for your population? For example, did you observe alterations of infants' behavior, physiology, development that result from exceeded sound levels?

With regard to your valuable input, we now added the lines 328ff: “When compared to other NICUs, our open bay NICU does not seem to be a loud one, the staff is trained to lower their voices, we offer music therapy and we encourage parents to communicate with their newborns. Depending on the child, some nurses observed that during the most common visiting hours such as during the weekends and daytimes, their newborns showed a higher rate of apneas or higher Finnegan-scores when pain reactions were quantified.

  • On page 8 you state: “Consequently, effort should be made to increase meaningful acoustic stimulation of preterm babies in an NICU, e.g., with Creative Music Therapy providing vibroacoustic enrichment by meaningful auditory stimulation and social contact with infant-directed music and parental empowerment to speak with “motherese” hum and sing for their infants [6,36].”

This is the first time you mention Music Therapy in the article, and the reader might not be familiar with it. Please explain briefly. You mention Creative Music Therapy as meaningful stimulation. Why do you specifically address CMT and not MT in general as meaningful stimulation? I would suggest justifying just mentioning CMT or leave it more general, since I believe this article is not specifically about CMT?

You are right that a detailed description of creative music therapy is beyond the scope of this article. Correspondingly we shortened the section you refer to line 365ff: . Consequently, effort should be made both to monitor and to decrease noise sources, for example via portable applications [23,24]. In addition, meaningful acoustic stimulation of preterm babies in a NICU should be increased providing vibroacoustic enrichment by meaningful auditory stimulation and social contact with infant-directed music and parental empowerment to speak and sing for their infants [18,25]. Offering zones where staff and/or parents can communicate and discuss with each other without disturbing the infants sleep is of great interest.

  • You state that specific aspects of the incubator (the fan, opening doors) exceed suggested sound levels. However, in the APA guidelines, peak sound levels for short periods of time are considered. You mention such guidelines briefly in the introduction section, but I would suggest coming back to them also in the discussion.

According to your suggestion, we now add the detailed recommendation again in the discussion section in line 311ff: “Unfortunately, these findings do not comply with the recommendations to maintain the combination of background and operational sound within an hourly equivalent continuous sound pressure level (SPL, Leq) of 50dB, referring to a A weighted slow response. Moreover, the SPL of 55dB should not be exceeded more than 10% of the time (l10). Opening the incubator at one port does not comply with the recommendation that transient sounds shall not exceed 70dB.”.

In the discussion you mention the WHO guidelines for community noise, which is probably not the most appropriate reference since the NICU is a specialized setting. Besides, the reference is from 1999 and is most likely not up to date. 

We tried to refer to the original source where the recommendation for environmental noise in NICUs has been formulated. There are updated standards, but they still seem to be based on the original source of the WHO. We consequently added the following sentence in line 184: “Since then, these standards have been repetitively integrated in the current recommendations for environmental standards in NICUs [26-29].

  • In the discussion there is a brief section on environmental noise in the NICU, but the references you mention are all from between 10 and 15 years ago. There are many new studies on the sound environment in the NICU, I would suggest updating your information.

We performed a structured review again and now integrated very recent publications in the discussion. In line 224 we added:  There is only one randomized controlled trial to reduce noise so far. However, in this small newborn cohort, silicone ear plugs may have been effective in the reduction of noise exposure [30], although they will not protect against respirator associated noise transferred via bone conduction and will not help in distinguishing noise from sound, either.

  • There is no section in the article that refers to a literature review. Besides that it is known that noise can alter preterm infants sleep, physiology and development, what are the sound levels in other units and countries? Here are just some examples:
    • doi: 0.1016/j.ijporl.2018.02.013 Zacarias
    • doi: 10.1111/apa.13906 Parra et al

We did a structured literature review to check whether the most important points are covered within this article. In addition, we can now offer to provide some qualitative data of a person rating all sounds while sitting inside the NICU.

  • In line 58 we add :  “Nevertheles,, sound pressure levels alone do not reflect whether a sound is perceived as comforting or disturbing- it might even be difficult to sleep in the presence of a buzzing mosquito, which usually does not elicit sound pressure levels higher than 40 dB(A) [31]. From a structured literature research using the databases Pubmed, EMBASE and Web of Science in July 2021, 500 reports on “noise and NICU” could be retrieved (Appendix 1). These publications demonstrate that noise impacts on newborns, parents and the staff [14,30,32-34]. They often either focus on the reduction of noise [30,35] or at the enrichment of the acoustic environment [36,37]. Many studies rely on spot measurements of sound pressure levels and do not consider any frequencies.”
  • I would suggest providing a more thorough literature review at the beginning of the article so the reader gets an idea of what is known already and can appreciate your study within a wider field of knowledge. Also, you should come back in the discussion and compare your results to other studies. What is unique about your study? How does your study help to further advance knowledge in the field? This should be addressed. There are no recommendations on what to do with the information obtained in your study besides referring to structural (NICU) and manufactural (incubator) improvements that should take place. What do your results mean for parents, staff, music therapists? What recommendation can you make? Should music therapists or parents open the doors of the incubators when talking or singing? Should they be closed? Are there any differences for voice vs instruments? How can staff use this information to better protect preterm babies? It is essential to extrapolate your findings to provide concrete information for relevant stakeholders in the NICU.

With the improvements named above, the structured literature review in the appendix and the optional qualitative assessment we could integrate we added considerable information to this article. In addition, we now refer to 43 additional publications in order to compare our results to other studies. The frequency response with fan in progress and assessed in different surroundings is unique and mentioned in the part “strengths”.

198ff: “Therefore, if a caregiver wants to read or sing to the child, the side port of the incubator should be opened. Additionally, the lower frequencies will be better audible than higher which might explain why in a recent analysis the exposure to mothers’ voices led inside an incubator induced less relaxing physiological reactions in the newborns compared to white noise exposure [38].”

227ff: “However, when lying in the warmer or the arms of the parents, alarms would still disturb the infants, families, and NICU staff, so that visual or sensory alarms might be an alternative option to reduce noise [39]. In order to ensure persistent noise reduction repetitive training and increased awareness of the personnel is very important, otherwise the effect of improvements will get lost [40].”

263ff: The assessment of what is comfortable and what is not for an individual patient can, therefore, only be based on close observation of the preterm. In contrast to adults where the sleeping time concentrates on several hours during the night, newborns have irregular sleeping pattern without an established circadian rhythm [41], which makes the organization of an open ward even more challenging. In order to adapt the acoustical environment to each patient’s needs, we would need a flexible acoustic environment in each patient zone, which could be realized with advanced acoustic curtains. Because family centered care increases interaction between parents and staff, the communication has to be properly dosed in order to provide rest and recreation for neighbouring families. A recent review on intensive care unit built environment addresses this problem [33]. Studies suggest that parental vocal qualities are commonly adapted to their infants’ behavioral state [42], that is why an attenuation of either the infants state and/or the parental voice by the noisy environment and/or incubator possibly impairs proper interaction. Current monitor alarms produce sound at or above 480 Hz, which is within the frequency range of newborn babies’ cries [43]. These characteristics are certainly useful to attract the staff’s attention, but they may impair newborn and parental comfort or rest.

286 ff: “The bit depth of 16 bit and a sampling rate of at least 48 kHz is far higher than in most studies where there were no continuous measurements.”

We added some recommendations in line 356 ff:

Both human factors (staff and families), direct (incubator) and indirect infant and personnel surroundings (building properties, drawers, wardrobes) as well as technical equipment (ventilators, monitoring) have to be considered. The staff and families need continuous education to reduce their sound pressure levels during interactive speech, but to increase the levels and open the incubator ports if the talk is directed to the newborns. While direct speech provides a very broad frequency range between 20 Hz and 20 kHz, the spectra of microphones and loud speakers are narrow (displayed by respective provider). Alarm levels should be reduced and incubator ports and drawers have to be opened gently. Some background noise will always be audible, but it will be interpreted differently depending on the infant’s health state and genetic background. Consequently, effort should be made both to monitor and to decrease noise sources, for example via portable applications [23,24]. In addition, meaningful acoustic stimulation of preterm babies in a NICU should be increased providing vibroacoustic enrichment by meaningful auditory stimulation and social contact with infant-directed music and parental empowerment to speak and sing for their infants [18,25]. Offering zones where staff and/or parents can communicate and discuss with each other without disturbing the infants sleep is of great interest

  1. O'Callaghan, N.; Dee, A.; Philip, R.K. Evidence-based design for neonatal units: a systematic review. Matern Health Neonatol Perinatol 2019, 5, 6, doi:10.1186/s40748-019-0101-0.
  2. van Veenendaal, N.R.; van Kempen, A.; Franck, L.S.; O'Brien, K.; Limpens, J.; van der Lee, J.H.; van Goudoever, J.B.; van der Schoor, S.R.D. Hospitalising preterm infants in single family rooms versus open bay units: A systematic review and meta-analysis of impact on parents. EClinicalMedicine 2020, 23, 100388, doi:10.1016/j.eclinm.2020.100388.
  3. Liu, W.F. Comparing sound measurements in the single-family room with open-unit design neonatal intensive care unit: the impact of equipment noise. J Perinatol 2012, 32, 368-373, doi:10.1038/jp.2011.103.
  4. Van Enk, R.A.; Steinberg, F. Comparison of private room with multiple-bed ward neonatal intensive care unit. HERD 2011, 5, 52-63, doi:10.1177/193758671100500105.
  5. Chen, H.L.; Chen, C.H.; Wu, C.C.; Huang, H.J.; Wang, T.M.; Hsu, C.C. The influence of neonatal intensive care unit design on sound level. Pediatr Neonatol 2009, 50, 270-274, doi:10.1016/S1875-9572(09)60076-0.
  6. Stevens, D.C.; Akram Khan, M.; Munson, D.P.; Reid, E.J.; Helseth, C.C.; Buggy, J. The impact of architectural design upon the environmental sound and light exposure of neonates who require intensive care: an evaluation of the Boekelheide Neonatal Intensive Care Nursery. J Perinatol 2007, 27 Suppl 2, S20-28, doi:10.1038/sj.jp.7211838.
  7. Stevens, D.C.; Helseth, C.C.; Thompson, P.A.; Pottala, J.V.; Khan, M.A.; Munson, D.P. A Comprehensive Comparison of Open-Bay and Single-Family-Room Neonatal Intensive Care Units at Sanford Children's Hospital. HERD 2012, 5, 23-39, doi:10.1177/193758671200500403.
  8. Stevens, D.; Thompson, P.; Helseth, C.; Pottala, J. Mounting evidence favoring single-family room neonatal intensive care. J Neonatal Perinatal Med 2015, 8, 177-178, doi:10.3233/NPM-15915035.
  9. Surenthiran, S.S.; Wilbraham, K.; May, J.; Chant, T.; Emmerson, A.J.; Newton, V.E. Noise levels within the ear and post-nasal space in neonates in intensive care. Arch Dis Child Fetal Neonatal Ed 2003, 88, F315-318, doi:10.1136/fn.88.4.f315.
  10. Hoehn, T.; Busch, A.; Krause, M.F. Comparison of noise levels caused by four different neonatal high-frequency ventilators. Intensive Care Med 2000, 26, 84-87, doi:10.1007/s001340050016.
  11. Kazemizadeh Gol, M.A.; Black, A.; Sidman, J. Bone conduction noise exposure via ventilators in the neonatal intensive care unit. Laryngoscope 2015, 125, 2388-2392, doi:10.1002/lary.25199.
  12. Cox, R.V.; Neto, S.F.D.C.; Lamblin, C.; Sherif, M.H. ITU-T coders for wideband, superwideband, and fullband speech communication [Series Editorial]. IEEE Communications Magazine 2009, 47, 106-109, doi:10.1109/MCOM.2009.5273816.
  13. Liu, C.; Fu, Q.J.; Narayanan, S.S. Effect of bandwidth extension to telephone speech recognition in cochlear implant users. J Acoust Soc Am 2009, 125, EL77-83, doi:10.1121/1.3062145.
  14. Elbaz, M.; Leger, D.; Sauvet, F.; Champigneulle, B.; Rio, S.; Strauss, M.; Chennaoui, M.; Guilleminault, C.; Mira, J.P. Sound level intensity severely disrupts sleep in ventilated ICU patients throughout a 24-h period: a preliminary 24-h study of sleep stages and associated sound levels. Ann Intensive Care 2017, 7, 25, doi:10.1186/s13613-017-0248-7.
  15. Herrmann, B.; Augereau, T.; Johnsrude, I.S. Neural Responses and Perceptual Sensitivity to Sound Depend on Sound-Level Statistics. Sci Rep 2020, 10, 9571, doi:10.1038/s41598-020-66715-1.
  16. Job, R.F.; Hatfield, J.; Carter, N.L.; Peploe, P.; Taylor, R.; Morrell, S. General scales of community reaction to noise (dissatisfaction and perceived affectedness) are more reliable than scales of annoyance. J Acoust Soc Am 2001, 110, 939-946, doi:10.1121/1.1385178.
  17. Pavlov, I.P. Conditioned reflexes: an investigation of the physiological activity of the cerebral cortex; Oxford Univ. Press: Oxford, England, 1927; pp. xv, 430-xv, 430.
  18. Haslbeck, F.B.; Jakab, A.; Held, U.; Bassler, D.; Bucher, H.U.; Hagmann, C. Creative music therapy to promote brain function and brain structure in preterm infants: A randomized controlled pilot study. Neuroimage Clin 2020, 25, 102171, doi:10.1016/j.nicl.2020.102171.
  19. Lim, N. Cultural differences in emotion: differences in emotional arousal level between the East and the West. Integr Med Res 2016, 5, 105-109, doi:10.1016/j.imr.2016.03.004.
  20. Kliuchko, M.; Heinonen-Guzejev, M.; Vuust, P.; Tervaniemi, M.; Brattico, E. A window into the brain mechanisms associated with noise sensitivity. Sci Rep 2016, 6, 39236, doi:10.1038/srep39236.
  21. Heinonen-Guzejev, M.; Vuorinen, H.S.; Mussalo-Rauhamaa, H.; Heikkila, K.; Koskenvuo, M.; Kaprio, J. Genetic component of noise sensitivity. Twin Res Hum Genet 2005, 8, 245-249, doi:10.1375/1832427054253112.
  22. McDermott, J.H.; Schultz, A.F.; Undurraga, E.A.; Godoy, R.A. Indifference to dissonance in native Amazonians reveals cultural variation in music perception. Nature 2016, 535, 547-550, doi:10.1038/nature18635.
  23. Gilmour, D.; Duong, K.M.; Gilmour, I.J.; Davies, M.W. NeoSTRESS: Study of Transfer and Retrieval Environmental StressorS upon neonates via a smartphone application - Sound. J Paediatr Child Health 2020, 56, 1396-1401, doi:10.1111/jpc.14947.
  24. Capriolo, C.; Viscardi, R.M.; Broderick, K.A.; Nassebeh, S.; Kochan, M.; Solanki, N.S.; Leung, J.C. Assessment of Neonatal Intensive Care Unit Sound Exposure Using a Smartphone Application. Am J Perinatol 2020, doi:10.1055/s-0040-1714679.
  25. Haslbeck, F.B.; Bassler, D. Music From the Very Beginning-A Neuroscience-Based Framework for Music as Therapy for Preterm Infants and Their Parents. Frontiers in behavioral neuroscience 2018, 12, 112, doi:10.3389/fnbeh.2018.00112.
  26. Noise: a hazard for the fetus and newborn. American Academy of Pediatrics. Committee on Environmental Health. Pediatrics 1997, 100, 724-727.
  27. Graven, S.N. Sound and the developing infant in the NICU: conclusions and recommendations for care. J Perinatol 2000, 20, S88-93, doi:10.1038/sj.jp.7200444.
  28. White, R.D. Recommended standards for the newborn ICU. J Perinatol 2007, 27 Suppl 2, S4-S19, doi:10.1038/sj.jp.7211837.
  29. White, R.D.; Consensus Committee on Recommended Design Standards for Advanced Neonatal, C. Recommended standards for newborn ICU design, 9th edition. J Perinatol 2020, 40, 2-4, doi:10.1038/s41372-020-0766-2.
  30. Almadhoob, A.; Ohlsson, A. Sound reduction management in the neonatal intensive care unit for preterm or very low birth weight infants. Cochrane Database Syst Rev 2020, 1, CD010333, doi:10.1002/14651858.CD010333.pub3.
  31. Mankin, R.W. Acoustical detection of Aedes taeniorhynchus swarms and emergence exoduses in remote salt marshes. J Am Mosq Control Assoc 1994, 10, 302-308.
  32. Thomas, K.A.; Martin, P.A. NICU sound environment and the potential problems for caregivers. J Perinatol 2000, 20, S94-99, doi:10.1038/sj.jp.7200435.
  33. Verderber, S.; Gray, S.; Suresh-Kumar, S.; Kercz, D.; Parshuram, C. Intensive Care Unit Built Environments: A Comprehensive Literature Review (2005-2020). HERD 2021, 19375867211009273, doi:10.1177/19375867211009273.
  34. Bry, A.; Wigert, H. Psychosocial support for parents of extremely preterm infants in neonatal intensive care: a qualitative interview study. BMC Psychol 2019, 7, 76, doi:10.1186/s40359-019-0354-4.
  35. Hutchinson, G.; Du, L.; Ahmad, K. Incubator-based Sound Attenuation: Active Noise Control In A Simulated Clinical Environment. PLoS One 2020, 15, e0235287, doi:10.1371/journal.pone.0235287.
  36. Pineda, R.; Guth, R.; Herring, A.; Reynolds, L.; Oberle, S.; Smith, J. Enhancing sensory experiences for very preterm infants in the NICU: an integrative review. J Perinatol 2017, 37, 323-332, doi:10.1038/jp.2016.179.
  37. Filippa, M.; Panza, C.; Ferrari, F.; Frassoldati, R.; Kuhn, P.; Balduzzi, S.; D'Amico, R. Systematic review of maternal voice interventions demonstrates increased stability in preterm infants. Acta Paediatr 2017, 106, 1220-1229, doi:10.1111/apa.13832.
  38. Liao, J.; Liu, G.; Xie, N.; Wang, S.; Wu, T.; Lin, Y.; Hu, R.; He, H.G. Mothers' voices and white noise on premature infants' physiological reactions in a neonatal intensive care unit: A multi-arm randomized controlled trial. Int J Nurs Stud 2021, 119, 103934, doi:10.1016/j.ijnurstu.2021.103934.
  39. Freudenthal, A.; van Stuijvenberg, M.; van Goudoever, J.B. A quiet NICU for improved infants’ health, development and well-being: a systems approach to reducing noise and auditory alarms. Cognition, Technology & Work 2012, 15, 329-345, doi:10.1007/s10111-012-0235-6.
  40. Casey, L.; Fucile, S.; Flavin, M.; Dow, K. A two-pronged approach to reduce noise levels in the neonatal intensive care unit. Early Hum Dev 2020, 146, 105073, doi:10.1016/j.earlhumdev.2020.105073.
  41. Wielek, T.; Del Giudice, R.; Lang, A.; Wislowska, M.; Ott, P.; Schabus, M. On the development of sleep states in the first weeks of life. PLoS One 2019, 14, e0224521, doi:10.1371/journal.pone.0224521.
  42. Saliba, S.; Esseily, R.; Filippa, M.; Gratier, M.; Grandjean, D. Changes in the vocal qualities of mothers and fathers are related to preterm infant's behavioural states. Acta Paediatr 2020, 109, 2271-2277, doi:10.1111/apa.15238.
  43. Shinya, Y.; Kawai, M.; Niwa, F.; Imafuku, M.; Myowa, M. Fundamental Frequency Variation of Neonatal Spontaneous Crying Predicts Language Acquisition in Preterm and Term Infants. Front Psychol 2017, 8, 2195, doi:10.3389/fpsyg.2017.02195.
